# ROADSIDE MONOCULAR 3D DETECTION VIA 2D-DETECTION PROMPTING

## ABSTRACT

The problem of roadside monocular 3D detection requires detecting objects of interested classes (e.g., vehicle and pedestrian) in a 2D RGB frame and predicting their 3D information such as Bird-Eye-View (BEV) locations. It has broad applications such as traffic control, vehicle-vehicle communication and vehicle-infrastructure cooperative perception. To approach this problem, we present a novel and simple method that significantly outperforms prior arts by *exploiting 2D detections to help 3D detections* based on two key insights. First, 2D detectors are much easier to train and perform significantly better than 3D detectors if measured on the 2D image plane. Second, plenty of publicly available 2D-box annotated datasets allows pretraining a strong base detector, which, once fine-tuned, yields a much better 2D detector for the roadside dataset. To exploit the 2D detector for 3D detection, we explore three techniques: (1) concatenating both 2D and 3D detectors' features, (2) prompting and attentively fusing 2D and 3D detectors' features, and (3) prompting and encoding predicted 2D boxes' $\{x, y,$ width, height, label$\}$ and attentively fusing such with the 3D detector's features. Surprisingly, the third performs significantly better than the others. We conjecture that prompting 2D detections gives pinpointed object targets for the 3D detector to learn how to inflate them to BEV as 3D detections. Moreover, we suggest a class-grouping strategy that merges classes based on their functionality, which leads to further improvements. Comprehensive ablation studies and extensive experiments demonstrate that our method achieves the state-of-the-art on two existing large-scale roadside 3D detection benchmarks.

## 1 INTRODUCTION

Roadside 3D detection is an emerging research problem (Yu et al., 2022; Ye et al., 2022) that has broad applications such as vehicle-vehicle communication (Chen et al., 2019; Li et al., 2022a), vehicle-infrastructure cooperative perception (Arnold et al., 2020), Intelligent traffic control (Rauch et al., 2012; Wang et al., 2020), etc. A practical yet challenging setup is to use a single RGB camera for roadside perception, known as *roadside monocular 3D detection* (Yang et al., 2023), which takes a monocular video frame as input and aims to detect objects of interested classes (e.g., vehicle and pedestrian), and predict their 3D information such as depth, 3D box and orientation.

**Status quo.** Roadside monocular 3D detection is an ill-posed problem because it requires inferring 3D information (e.g., depth, 3D shape and orientation) from 2D cues. Perhaps fortunately, camera pose is fixed on roadside infrastructure, allowing training (deep neural) models on massive annotated data to learn to infer such 3D information. To foster the research of roadside monocular 3D detection, the community has recently released two large-scale benchmark datasets, DAIR-V2X-I (Yu et al., 2022) and Rope3D (Ye et al., 2022), and tried to train end-to-end deep neural networks for 3D detection (Li et al., 2022b; 2023). However, depth estimation for far objects remains rather difficult (Welchman et al., 2004; Rushton & Duke, 2009; Gupta et al., 2023). To mitigate this difficulty, Yang et al. (2023) proposes BEVHeight that converts the problem of depth estimation to object height estimation and exploits camera parameters to infer object depth, achieving the state-of-the-art performance of roadside monocular 3D detection on both DAIR-V2X-I and Rope3D.

**Motivation.** We empirically show that the state-of-the-art 3D detector BEVHeight underperforms the 2D detector DINO (Zhang et al., 2023) in terms of 2D detection performance, as quantitatively

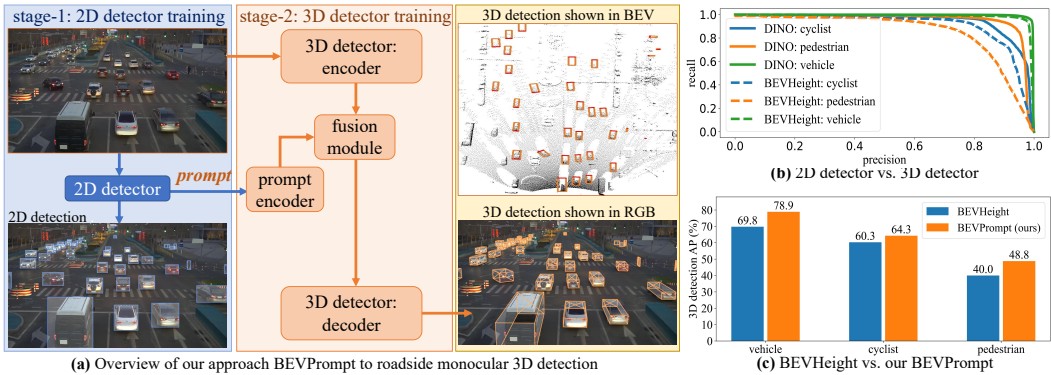

Figure 1: **(a)** To approach roadside monocular 3D detection, we introduce BEVPrompt, which first trains a 2D detector and then exploits it to train the monocular 3D detector which has two crucial modules: *prompt encoder* and *fusion module*. **(b)** This is motivated by the observation that, a simple 2D detector resoundingly outperforms the state-of-the-art 3D detector BEVHeight (Yang et al., 2023) in terms of 2D detection performance on DAIR-V2X-I dataset. This indicates that 2D detection is an "easier" task compared to monocular 3D detection, even though we use the same dataset to train them. We explore how to exploit the 2D detector to facilitate 3D detector training, e.g., prompting 2D detections and attentively fusing them with 3D detector's features (details in Fig. 2). **(c)** Our BEVPrompt significantly outperforms the state-of-the-art method BEVHeight (cf. Table 1).

demonstrated by precision-recall curves on the DAIR-V2X-I dataset in Fig. 1b. To be specific, to train DINO, we map ground-truth 3D annotations to image plane to derive 2D box annotations; to evaluate BEVHeight w.r.t 2D detections, we project its 3D detections on the 2D image plane. Clearly, DINO achieves significantly better performance than BEVHeight on 2D object detection. While this comparison seems unfair because DINO and BEVHeight have different network architectures, it still has several implications: (1) 2D detection is a much "easier" task than monocular 3D detection, (2) 2D detectors are easier to train than 3D detectors, (3) 2D detectors are better understood and their techniques are better explored than 3D detectors, etc. This comparison thus motivates us to *improve roadside monocular 3D detection by exploiting a 2D detector*. Intuitively, when a well-trained 2D detector produces high-quality 2D detections, the problem of 3D detection boils down to inflating the 2D detections to BEV.

**Technical insights.** We study how to exploit a 2D detector to help train better roadside monocular 3D detectors. Loosely speaking, directly training 3D detectors involves combining multiple loss terms, such as 2D box coordinate regression loss, depth estimation loss, orientation regression loss, classification loss, etc. This loss aggregation can complicate the training process and might lead to the missing of some hard cases as an attempt to handle the difficulty in regressing coordinates, depth and orientation. In contrast, training a 2D detector is simpler that incorporates fewer loss terms (e.g., 2D box coordinate regression loss and classification loss) and is well studied in the literature. Furthermore, there is plenty of publicly available datasets of 2D object detection, allowing pretraining a strong base 2D detector and finetuning it on the target dataset of roadside scenes. Our techniques study how to effectively prompt[1] what is learned in a 2D detector to help train better 3D detectors (Fig. 1a). A straightforward method is to concatenate the *features* computed by the 2D detector with the 3D detector's features. This notably improves the 3D detection performance. A further approach is to attentively fuse these *features*, inspired by recent transformer architectures or attention modules, which performs notably better than the simple concatenation (Table 5). Lastly, a rather simple but effective method we present is to prompt 2D *detection output* (i.e., 2D boxes' coordinates and class labels) to attentively fuse 3D detector's features, leading to significant improvement over the state-of-the-art (Fig. 1c). We conjecture that using 2D detections gives pinpointed targets, based on which training the 3D detector boils down to a simpler problem of inflating 2D detections to BEV.

**Technical contributions.** We reiterate our technical contribution as the proposed BEVPrompt method (Fig. 1a), which exploits a 2D detector trained on the same roadside dataset for 3D detection. We study multiple approaches to exploit the 2D detector in a way of prompting either its features or outputs. Our final method prompts the 2D detection output as a 5-dim vector $\{x,$

---

[1]We use "prompt" to emphasize that the information used to help train 3D detectors is provided by the 2D detector, which is frozen during training 3D detectors.

$y$, width, height, label}, encodes it and attentively fuses it with features of the 3D detector being trained. Moreover, we suggest a class-grouping strategy to train multiple heads in the 3D detector, leading to further improvement. Comprehensive ablation studies and extensive experiments demonstrate that our BEVPrompt resoundingly outperforms the state-of-the-art on the two existing large-scale benchmark datasets.

## 2 RELATED WORK

**Roadside monocular 3D detection** is proposed to help autonomous driving. The motivation is that autonomous vehicles have limited range of perception (Gupta et al., 2023), but a camera hung on a high infrastructure provides perception in a longer range (Yang et al., 2023). Through vehicle-infrastructure cooperative perception, vehicles can drive more safely. In addition, roadside monocular 3D detection can also help unexpected events detection and traffic congestion prediction. To foster research of 3D object detection on roadside scenes, the recent literature established two large-scale datasets, DAIR-V2X-I (Yu et al., 2022) and Rope3D (Ye et al., 2022). Among several recent approaches to roadside 3D monocular detection, BEVHeight (Yang et al., 2023) achieves the state-of-the-art performance on the two datasets. It proposes to train a model to regress towards objects height instead of object distance or depth, as done previously (Li et al., 2023). It finds that regressing height is an easier task than regressing depth given the fixed camera pose. Our work recognizes that a fruitful space to improve over BEVHeight is to train a 2D detector (Fig. 1b) using 2D box annotations derived by projecting the 3D annotations on the image plane and then exploit the 2D detector to facilitate 3D detector training (Fig. 1a). We show that our method significantly outperforms BEVHeight (Fig. 1c) on the two well-established datasets.

**2D object detection** is a fundamental computer vision problem (Felzenszwalb et al., 2009; Lin et al., 2014a), requiring detecting all objects belonging to some predefined categories. Deep learning approaches greatly improved 2D detection performance in the past years, primarily through end-to-end training a convolutional neural network (CNN) as the detector backbone and a detector-head for box regression and classification (Ren et al., 2015; Redmon et al., 2016; Liu et al., 2016; Wang et al., 2022). Recent developments in transformer architectures improve 2D object detection significantly further (Carion et al., 2020a; Zhu et al., 2020; Zhang et al., 2023). We explore how to leverage a 2D detector to train better monocular 3D detectors on roadside scenes. We simply use a state-of-the-art 2D detector DINO (Zhang et al., 2023) in this work. It is important to note that any 2D detectors can be used in our framework. To the best of our knowledge, we make the first attempt to explicitly exploit a 2D detector to improve roadside monocular 3D detection.

**Prompting** enabled by large language models (LLMs) is first recognized in the community of Natural Language Processing (NLP) (Brown et al., 2020; Sanh et al., 2022; Wei et al., 2022). Briefly, given a generative LLM, using a text prompt can instruct the model to produce desired text output. Inspired by this, the computer vision community endeavors to train large visual/multimodal models that can take text prompts to generate, restore and edit images (Ramesh et al., 2021; Liu et al., 2023; Nichol et al., 2021). Some recent works use visual image as prompts to guide visual models to accomplish vision tasks such as segmentation and inpainting (Bar et al., 2022; Wang et al., 2023). Beyond visual and text prompts, recent work prompts more diverse information to guide models to produce desired output. For example, the Segment Anything Model (SAM) accepts bounding boxes, segmentation masks, and foreground/background points as prompts towards interactive segmentation (Kirillov et al., 2023). Loosely speaking, SAM uses prompts as conditions, i.e., make segmentation predictions conditioned on user input. Inspired by above, we train a 3D detector that accepts prompts given by a well-trained 2D detector. The prompt can be as simple as predicted 2D coordinates and labels, similar to what used in SAM. Our method of prompting 2D detections for 3D detection significantly improves performance (Fig. 1c, Table 1).

## 3 ROADSIDE MONOCULAR 3D DETECTION VIA 2D-DETECTION PROMPTING

We first present the problem of roadside monocular 3D object detection, then introduce our approach, its important modules along with design choices, and techniques for further improvements.

**Problem definition.** Roadside monocular 3D detection requires detecting objects from a 2D RGB image and predicting their 3D information including $(x, y, z, w, h, l, yaw)$, where $(x, y, z)$ is the

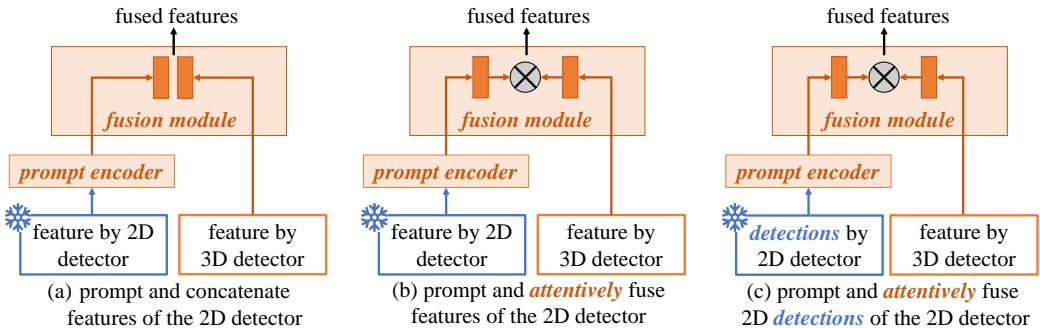

Figure 2: We study three designs of *prompt encoder* and *fusion module* (Fig. 1a), used to exploiting the 2D detector to facilitate 3D detector training. Design **(a)** concatenates feature maps extracted by the 2D detector and the 3D detector encoder. Design **(b)** extracts a feature vector based on a 2D detection's coordinates, encodes it through a *prompt encoder* (implemented as a MLP), and then attentively fuses them with features extracted by the 3D detector encoder through the *fusion module* (implemented by a transformer building block). Design **(c)** prompts and encodes a 2D detection as a 5-dim vector (coordinates $x$ and $y$, object width $w$ and height $h$, predicted class label), and attentively fuses the encoded feature with the feature maps of the 3D detector encoder through the fusion module, which is implemented by a transformer.

object's center location in the 3D world, $(w, h, l)$ denotes the width, height and length of the cuboid capturing the 3D object, and $yaw$ denotes the orientation of the cuboid on the BEV plane.

**Remarks.** Directly training the 3D detector must sum up multiple loss terms, including classification loss, depth estimation loss, 3D coordinate regression loss, orientation estimation loss, optionally velocity regression loss, etc. In contrast, 2D detection is a simpler task that requires detecting objects and predicting their class labels and 2D box coordinates on the image plane. As a result, training 2D detectors incorporates fewer loss terms: a classification loss and a 2D coordinate regression loss. If measured by 2D detection performance on the image plane, 2D detectors resoundingly outperform the 3D counterpart as convincingly demonstrated by Fig. 1b and Table 4. This motivates us to exploit a well-trained 2D detector to improve the training of 3D detectors.

### 3.1 THE PROPOSED METHOD: BEVPROMPT

Our BEVPrompt makes use of a well-trained 2D detector for 3D detection (Fig. 1a). We first train a 2D detector using the *same* training set consisting of monocular images and their 3D annotations on the objects of interest. We use the camera extrinsic and intrinsic parameters to project the 3D annotations onto the 2D image plane as 2D box annotations. After training the 2D detector, we freeze it and "prompt" information extracted by it (e.g., features or 2D detections) to train the 3D detector, which has an encoder, a decoder for 3D prediction, and importantly, two novel modules: *prompt encoder* and *fusion module*. The former encodes the prompted information given by the 2D detector into features, and the latter fuses them with 3D detector's features (given by its encoder). We explain the two modules below.

**Prompt encoder** intends to encode information provided by the 2D detector into features, which are prompted to help the 3D detector make 3D detections. The prompt encoder can be implemented as either a set of convolutional layers or a multi-layer perceptron depending on what is prompted and how to fuse (explained in the *fusion module* below). For example, if feature maps of the 2D detector are prompted, then the prompt encoder can be a set of convolutional layers with non-linear units (e.g., ReLU) that transform them into new feature maps at appropriate dimensions (Fig. 2a). This ensures that such new features can be fused with those of the encoder in the 3D detector through the *fusion module* explained below. In our work, we consider three different features to prompt: (1) feature maps represented as a tensor, (2) feature vectors at the spatial position specified by the predicted coordinates of a 2D detection, and (3) the 2D detection output (e.g., box coordinates $x$ and $y$, width, height, and class label). Somewhat surprisingly, (3) performs the best! To encode a 2D detection, we represent a predicted 2D box with a normalized positional encoding of its top-left and bottom-right corners, and represent the class label as a class index ID. We learn a multi-layer perceptron to transform theem into a feature vector (cf. details in appendix A).

**Fusion module** intends to fuse the prompted features (from the prompt encoder) and feature maps of the 3D detector encoder (Fig. 2). If the prompt features are feature maps, then a straightforward fusion method is to concatenate them (Fig. 2a), as widely adopted in the literature (Long et al., 2015; Pinheiro et al., 2016). Inspired by the recent transformer architecture (Vaswani et al., 2017) and its application in encoding prompts (Kirillov et al., 2023), we propose to use an attention module to fuse these features. Experiments demonstrate that such attentive fusion outperforms concatenation (Table 5). Concretely, we denote the feature maps from the encoder $F$ in the 3D detector for an input image $X$ as $F(X)$; the prompt encoder $F_{prompt}$ transforms the input prompt $p$ into feature $F_{prompt}(p)$. The attentively fused feature $f$ is computed below:

$$f = Transformer\Big(F(X), \ F_{prompt}(p)\Big) \tag{1}$$

The *Transformer* does the following (cf. details in appendix A): (1) applying self-attention on the prompted feature $F_{prompt}(p)$, producing a new feature $f_1$; (2) applying cross-attention between $f_1$ as query and $F(X)$, followed by an MLP that produces feature $f_2$; (3) applying cross-attention between $F(X)$ as query and $f_2$, producing the fused feature $f$.

### 3.2 DISCUSSIONS AND REMARKS

Our BEVPrompt exploits the output of a 2D detector for 3D detection (Fig. 1a). This allows our method to incorporate any 2D detectors and benefit from them to improve 3D detection. We make two remarks below.

**Techniques of 2D detection are well established.** In object detection, 2D detection is better explored than monocular 3D detection, primarily due to historical reasons that the former has been a fundamental problem in computer vision whereas the latter is relatively new. There are many excellent 2D detectors developed in the past years along with the developments of CNNs and transformers, such as FasterRCNN (Ren et al., 2015), YOLO (Bochkovskiy et al., 2020; Wang et al., 2022), DETR (Carion et al., 2020b) and DINO (Zhang et al., 2023). In this work, we use the transformer-based detector DINO (Zhang et al., 2023), which achieves the state-of-the-art performance on various 2D detection benchmark datasets.

**More datasets of 2D detection are publicly available.** As 2D detection is a fundamental problem in computer vision and has been actively explored since the past decade, many large-scale datasets are publicly available. In contrast, for the relatively new problem of monocular 3D detection, there are less datasets which are also small in scale compared to the 2D counterpart. For example, the 2D detection dataset COCO (Lin et al., 2014b) was published in 2014 and annotated 330K images with 2D boxes for 80 classes, while the 3D detection dataset nuScenes (Caesar et al., 2020) was published in 2020 and annotated 144K RGB images with 3D cuboids for 23 classes. One major reason is that labeling objects on images using 2D boxes is significantly cheaper and easier than labeling 3D cuboids on images. Note that labeling 3D cuboids often relies on other sensors such as LiDAR so to obtain 3D information such as depth for 3D annotation. The availability of large-scale 2D detection datasets allows pretraining a powerful 2D detector, using which facilitates 3D detector training with our framework (Fig. 2a).

### 3.3 FURTHER IMPROVEMENT BY CLASS GROUPING AND MULTI-HEAD DESIGN

Class grouping merges classes into superclasses and facilitates training by allowing features to be shared among related classes. Multi-head design resembles expert models which share feature backbone. Class grouping and multi-head design are usually used jointly (Yin et al., 2021) such as a head makes predictions only for specific classes that are grouped together. BEVHeight (Yang et al., 2023) adopts them considering the similarity of object appearance, e.g., size and shape in the 3D world. For example, on DAIR-V2X-I, BEVHeight creates three superclasses that merge {*truck, bus*}, {*car, van*}, and {*bicyclist, tricyclist, motorcyclist, barrowlist*}, respectively, along with the origin *pedestrian* class. It learns a four-head detector with each head making predictions for the corresponding superclass. We call this grouping strategy *appearance-based grouping*. Differently, we adopt *functionality-based grouping*, i.e., merging {*car, van, truck, bus*} into *vehicle*, and {*bicyclist, tricyclist, motorcyclist, barrowlist*} into *cyclist*. This is motivated by the fact that vehicles, cyclists and pedestrians appear at relatively fixed regions on the image. We expect this grouping strategy to facilitate training object detectors. Perhaps coincidentally, the DAIR-V2X-I benchmark also merges

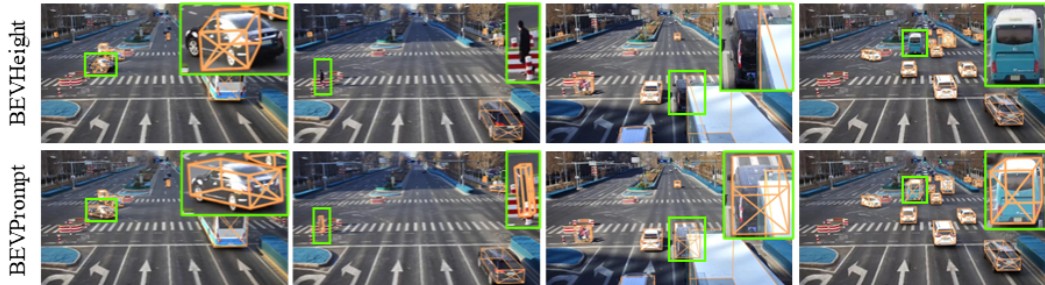

Figure 3: Visual comparison between the state-of-the-art method BEVHeight (Yang et al., 2023) and our BEVPrompt. Results show that our BEVPrompt can (1) make better orientation predictions than BEVHeight (column-1), and (2) detect objects missed by BEVHeight due to heavy small size (column-2), occlusion (column-3) and long distance (column-4).

Table 1: **Benchmarking results on the DAIR-V2X-I validation set.** Following Yang et al. (2023), we report numbers using the AP metric in bird-eye-view at IoU=0.5, 0.25, 0.25 on the three superclasses (Vehicle, Cyclist, and Pedestrian), respectively. As our framework BEVPrompt can use any detector achitectures, we build it using BEVDepth or BEVHeight. Results show that incorporating them in BEVPrompt significantly outperforms the prior methods on all classes. Notably, our approach outperforms the original BEVHeight by 8.8% in AP on Pedestrian! This attributes to the 2D detector which detects pedestrians much better than 3D detectors, cf. Table 4. More results on this dataset w.r.t other metrics are in Table 3 and 4.

| Method | MM | Vehicle$_{(IoU=0.5)}$ | | | Cyclist$_{(IoU=0.25)}$ | | | Pedestrian$_{(IoU=0.25)}$ | | |
|---|---|---|---|---|---|---|---|---|---|---|
| | | Easy | Mid | Hard | Easy | Mid | Hard | Easy | Mid | Hard |
| PointPillars (Lang et al., 2019) | L | 63.1 | 54.0 | 54.0 | 38.5 | 22.6 | 22.5 | 38.5 | 37.2 | 37.3 |
| SECOND (Yan et al., 2018) | L | 71.5 | 54.0 | 54.0 | 54.7 | 31.1 | 31.2 | 55.2 | 52.5 | 52.5 |
| MVXNet (Sindagi et al., 2019) | L + C | 71.0 | 53.7 | 53.8 | 54.1 | 30.8 | 31.1 | 55.8 | 54.5 | 54.4 |
| ImvoxelNet (Rukhovich et al., 2022) | C | 44.8 | 37.6 | 37.6 | 21.1 | 13.6 | 13.2 | 6.8 | 6.7 | 6.7 |
| BEVFormer (Li et al., 2022b) | C | 61.4 | 50.7 | 50.7 | 22.2 | 22.1 | 22.1 | 16.9 | 15.8 | 16.0 |
| BEVDepth (Li et al., 2023) | C | 75.5 | 63.6 | 63.7 | 55.7 | 55.5 | 55.3 | 35.0 | 33.4 | 33.3 |
| BEVHeight (Yang et al., 2023) | C | 77.8 | 65.8 | 65.9 | 60.2 | 60.1 | 60.5 | 41.2 | 39.3 | 39.5 |
| **BEVPrompt (Ours)** w/ BEVDepth | C | 77.1 | 69.2 | 69.1 | 57.1 | 57.0 | 56.9 | 41.3 | 40.0 | 40.2 |
| **BEVPrompt (Ours)** w/ BEVHeight | C | **82.4** | **77.2** | **77.3** | **62.0** | **64.9** | **65.2** | **50.2** | **48.1** | **48.2** |

classes into such superclasses for evaluation, presumably aiming to treat classes within each super-class equally for roadside surveillance and vehicle-infrastructure cooperative perception. Indeed, our class grouping strategy remarkably improves detection performance (Table 6).

# 4 EXPERIMENTS

We validate our methods through extensive experiments. First, we present experimental settings including two benchmark datasets, evaluation metrics and implementations details. Then, we compare our BEVPrompt against prior works including BEVHeight (Yang et al., 2023) which has achieved the best numbers on the benchmarks (till now). Lastly, we present ablation study for our method.

## 4.1 EXPERIMENTAL SETTINGS

**Datasets.** We use DAIR-V2X-I (Yu et al., 2022) and Rope3D (Ye et al., 2022) in experiments. Both datasets are publicly available for academic purposes. They contain LiDAR data but monocular 3D detection does not use it; LiDAR can be though of as a tool for 3D annotation and visualization.

- DAIR-V2X-I is the first large-scale dataset supporting the research of roadside monocular 3D detection. It has 10k images which are split into training (50%), validation (20%), and testing (30%) sets. As the testing set is not publicly available, we follow Yang et al. (2023) to report benchmarking results on the validation set, with breakdown analysis w.r.t three superclasses, "vehicle", "cyclist", and "pedestrian", at three difficulty levels (Easy, Medium, Hard).

- Rope3D is another large-scale roadside dataset that contains 50k images and annotates over 1.5M 3D objects in various scenes. Following Yang et al. (2023), we use its training set (70%) to train models and the validation set (30%) to report benchmarking results.

Table 2: **Benchmarking results on the Rope3D val-set.** We follow Ye et al. (2022) to use metrics AP and Rope at IoU=0.5 and 0.7. We build our BEVPrompt with BEVHeight, improving the latter significantly.

| Method | IoU = 0.5 | | | | IoU = 0.7 | | | |
| | Car | | Big Vehicle | | Car | | Big Vehicle | |
| | AP | Rope | AP | Rope | AP | Rope | AP | Rope |
|---|---|---|---|---|---|---|---|---|
| M3D-RPN (Brazil & Liu, 2019) | 54.2 | 62.7 | 33.1 | 44.9 | 16.8 | 32.9 | 6.9 | 24.2 |
| Kinematic3D (Brazil et al., 2020) | 50.6 | 58.9 | 37.6 | 48.1 | 17.7 | 32.9 | 6.1 | 22.9 |
| MonoDLE (Ma et al., 2021) | 51.7 | 60.4 | 40.3 | 50.1 | 17.7 | 32.9 | 6.1 | 22.9 |
| MonoFlex (Zhang et al., 2021) | 60.3 | 66.9 | 37.3 | 48.0 | 33.8 | 36.1 | 10.1 | 26.2 |
| BEVFormer (Li et al., 2022b) | 50.6 | 58.8 | 34.6 | 45.2 | 24.6 | 38.7 | 10.1 | 25.6 |
| BEVDepth (Li et al., 2023) | 69.6 | 74.7 | 45.0 | 54.6 | 42.6 | 53.1 | 21.5 | 35.8 |
| BEVHeight (Yang et al., 2023) | 74.6 | 78.7 | 48.9 | 57.7 | 45.7 | 55.6 | 23.1 | 37.0 |
| **BEVPrompt (Ours) w/ BEVHeight** | **82.2** | **84.7** | **73.9** | **77.4** | **49.1** | **58.3** | **30.7** | **42.9** |

Table 3: **State-of-the-art comparison on the DAIR-V2X-I validation set w.r.t 3D mAP**, which averages mAP with 3D cuboid IoU with thresholds from 0.5 to 0.95 and stepsize 0.05. This threshold range is used in other well-established benchmarks such as COCO (Lin et al., 2014b). We use BEVHeight and BEVDepth as the state-of-the-art methods (cf. Table 1). Our approach BEVPrompt resoundingly outperforms BEVHeight and BEVDepth on all the three classes. More results on this dataset w.r.t other metrics are in Table 4.

| Method | Vehicle$_{(3D\ IoU=0.5:0.95)}$ | | | Cyclist$_{(3D\ IoU=0.5:0.95)}$ | | | Pedestrain$_{(3D\ IoU=0.5:0.95)}$ | | |
| | Easy | Mid | Hard | Easy | Mid | Hard | Easy | Mid | Hard |
|---|---|---|---|---|---|---|---|---|---|
| BEVDepth (Li et al., 2023) | 36.3 | 36.1 | 36.0 | 3.3 | 3.4 | 3.4 | 1.1 | 1.0 | 1.0 |
| BEVHeight (Yang et al., 2023) | 37.4 | 37.2 | 37.2 | 4.7 | 4.9 | 5.0 | 1.7 | 1.7 | 1.7 |
| **BEVPrompt (Ours) w/ BEVDepth** | 40.8 | 41.2 | 41.5 | 5.3 | 5.0 | 5.3 | 1.9 | 1.8 | 1.8 |
| **BEVPrompt (Ours) w/ BEVHeight** | **43.8** | **44.0** | **44.1** | **7.1** | **6.8** | **7.0** | **3.0** | **3.0** | **3.0** |

**Metrics.** The literature of roadside monocular 3D detection and general object detection uses various metrics to benchmark methods. As using a single metric is insufficient, we use multiple ones that are well-established in the literature, as summarized below.

- $AP_{IoU=t}$ is the (average) precision (AP) at IoU threshold $t$. Yang et al. (2023) set $t = 0.5$ when evaluating on cyclist and pedestrian classes, and $t = 0.25$ on vehicle, on the BEV plane. Yet, these IoU thresholds are too relaxing w.r.t today's standards, e.g., the literature of object detection (Lin et al., 2014b) uses mean AP (mAP) which first computes per-class precision averaged at IoU thresholds from 0.5 to 0.95 with a step-size 0.05, and reports the mean of per-class APs. We additionally use mAP in our work explained next.

- The mAP considers various IoU thresholds and all the (imbalanced) classes. We report mAP on both BEV plane and image plane (as done in object detection literature) to benchmark methods. We further extend mAP with cuboid IoU averaged over 0.5 to 0.95 to benchmark 3D detection performance in the real 3D world, denoted by "3D mAP" (e.g., in Table 3).

- On the Rope3D benchmark dataset (Ye et al., 2022), we report numbers w.r.t its official metrics of $AP_{IoU=t}$ in BEV (Simonelli et al., 2019) and Rope$_{score}$. The latter is a sophisticated metric that jointly considers detection errors of orientation, area, and BEV box coordinates compared to the ground-truth.

**Implementations.** We design the monocular 3D detector by following (Yang et al., 2023), i.e., implementing the image encoder with ResNet101 (He et al., 2016), the proposal detection module with an FPN (Yan et al., 2018), the BEV encoder with PointPillar (Lang et al., 2019), and detector head as done in CenterPoint (Yin et al., 2021). We build our BEVPrompt atop of either BEVDepth or BEVHeight, allowing fair comparison against them. We train DINO (Zhang et al., 2023) as the 2D detector using the default hyperparameter setting. For all the methods, the input image resolution is 864x1536 and the initial voxel resolution is 1024x1024x1. We use random scaling and rotation to augment training images. We train all models for 50 epochs using AdamW optimizer (Loshchilov & Hutter, 2017) with learning rate 8e-4. We select the best checkpoint via validation. In post-processing, we remove low-scoring detections using a confidence threshold 0.3.

## 4.2 BENCHMARKING RESULTS

We compare our BEVPrompt against prior works including the previously best performing BEVHeight (Yang et al., 2023) on DAIR-V2X-I (Table 1) and Rope3D (Table 2). Results demonstrate that our BEVPrompt achieves the state-of-the-art, resoundingly outperforming prior methods. Fig. 3 compares visual results by BEVPrompt and BEVHeight (see more visualizations in appendix

Table 4: **State-of-the-art comparison on the DAIR-V2X-I validation set w.r.t 2D mAP on the image plane.** In object detection, 2D mAP is wide used and we list results w.r.t this metric to demonstrate that a 2D detector performs significantly better than 3D detectors in terms of detecting objects on RGB images if measured on the image plane. We report 2D mAP for 3D detectors by projecting their 3D detections on the image plane so derive their predicted 2D boxes. We also list results of a 2D detector DINO trained on the training set of DAIR-V2X-I, which achieves >10 mAP than the state-of-the-art detector BEVHeight on all classes. In particular, DINO is nearly 20 mAP higher than BEVHeight on the pedestrian class! Owing to the power of 2D detection, our method BEVPrompt significantly outperforms BEVHeight, for both of which we project their 3D detections to 2D image plane for evaluation.

| Method | Vehicle$_{(2D\ IoU=0.5:0.95)}$ | | | Cyclist$_{(2D\ IoU=0.5:0.95)}$ | | | Pedestrian$_{(2D\ IoU=0.5:0.95)}$ | | |
|---|---|---|---|---|---|---|---|---|---|
| | Easy | Mid | Hard | Easy | Mid | Hard | Easy | Mid | Hard |
| 2D Detector (DINO) (Zhang et al., 2023) | 74.3 | 75.6 | 75.6 | 57.3 | 54.7 | 54.9 | 50.6 | 51.6 | 51.6 |
| BEVDepth (Li et al., 2023) | 62.9 | 64.8 | 64.2 | 41.5 | 44.8 | 44.9 | 32.9 | 33.1 | 33.0 |
| BEVHeight (Yang et al., 2023) | 63.5 | 64.7 | 64.5 | 41.4 | 44.6 | 44.7 | 33.6 | 33.7 | 33.9 |
| **BEVPrompt (Ours)** w/ BEVDepth | 68.0 | 68.3 | 68.9 | 42.9 | 47.7 | 49.6 | 49.1 | 42.5 | 43.6 |
| **BEVPrompt (Ours)** w/ BEVHeight | 68.1 | 68.8 | 68.8 | 43.5 | 47.9 | 49.5 | 49.4 | 42.8 | 43.5 |

Table 5: **Ablation on what to prompt from the 2D detector.** We compare four prompting methods in our framework against the state-of-the-art BEVHeight. Prompting either features or detection output remarkably improves the final 3D detection performance. Surprisingly, prompting the simplistic box coordinates of 2D detections outperforms prompting features. Additionally prompting predicted class labels brings further improvement.

| Method | Vehicle$_{(IoU=0.7)}$ | | | Cyclist$_{(IoU=0.5)}$ | | | Pedestrian$_{(IoU=0.5)}$ | | |
|---|---|---|---|---|---|---|---|---|---|
| | Easy | Mid | Hard | Easy | Mid | Hard | Easy | Mid | Hard |
| BEVHeight | 67.4 | 67.2 | 67.2 | 23.6 | 25.9 | 26.5 | 11.5 | 11.4 | 11.5 |
| concatenating feature maps | 70.6 | 70.3 | 70.4 | 24.9 | 27.5 | 28.0 | 11.6 | 11.4 | 11.4 |
| attentively fusing feature vectors | 70.7 | 71.2 | 71.5 | 25.0 | 27.7 | 28.2 | 12.5 | 12.1 | 12.4 |
| attentively fusing 2D box coordinates | 72.6 | 74.0 | 71.8 | 25.7 | 28.2 | 28.8 | 13.8 | 13.6 | 13.6 |
| attentively fusing 2D box coordinates and labels | **72.8** | **74.2** | **72.5** | **27.7** | **31.5** | **31.8** | **15.3** | **15.2** | **15.2** |

Fig. 7). Table 3 lists comparisons w.r.t the 3D mAP metric on DAIR-V2X-I. As the prior methods BEVDepth and BEVHeight report the best results previously, we built our BEVPrompt on top of them. Results demonstrate that our BEVPrompt still outperforms them remarkably.

We further benchmark methods w.r.t 2D detection performance. This helps us understand that the 2D detection performance greatly affects the 3D detection performance, justifying our motivation of leveraging a well-trained 2D detector to improve 3D detector. To report 2D mAP metrics for 3D detectors including our BEVPrompt, we project 3D detections onto the image plane and compare against ground-truth 2D boxes. Table 4 compares these methods, as well as the 2D detector DINO which is exclusively trained for 2D detection. As expected, DINO achieves the best. Exploiting DINO, our method BEVPrompt improves the 2D detection performance of BEVDepth and BEVHeight when built on them.

## 4.3 Ablation Study

**What to prompt from the 2D detector.** To exploit the 2D detector to help train 3D detectors, we study what to prompt, including its intermediate feature maps, feature vector located at a 2D detector, and even the simplistic 2D detection output (i.e., predicted box coordinates and class label). Note that prompting different things requires different fusion methods. For example, when prompting feature maps, one can concatenate them with features of the encoder in the 3D detector, while prompting a feature vector or detection output require fusing a vector with the feature maps. Therefore, we use concatenation to fuse the prompted feature maps, and an attention module to fuse the latter. Table 5 lists results. Perhaps surprisingly, prompting the simple detection coordinates performs better than feature maps and feature vectors located at 2D detections, achieving [71.8, 28.8, 13.6] AP for `Hard` examples of the three classes [vehicle, cyclist, pedestrian], respectively. Furthermore, additionally prompting the predicted class label performs the best performance, boosting performance to [72.5, 31.8, 15.2] from the previous results.

**How to group classes and design the 3D detector head.** Section 3.3 details class grouping strategies and multi-head design. BEVHeight adopts an *appearance-based grouping* (Yang et al., 2023), e.g., mering big vehicle classes bus and truck together; whereas we propose *functionality-based grouping*, e.g., merging all vehicles together including bus, truck, van, and car. It is worth noting that benchmarking metrics also evaluate on such "functionality-grouped" superclasses. Table 6 com-

Table 6: **Ablation on class-grouping strategies.** Grouping classes differently with multi-head design (each detector head makes predictions within a specific superclass) makes a difference in training and yields detectors performing differently. Briefly, instead of grouping classes according to their object appearance (as done by BEVHeight), we group them according to their functionalities, e.g., merging {car, van, truck, bus} into the superclass vehicle. Note that these superclasses are exactly what are evaluated on in the DAIR-V2X-I benchmark protocol. We replace BEVHeight's original appearance-based grouping strategy with our functionality-based grouping, achieving notable improvement (cf. the first two rows). Using our functionality-based grouping strategy in BEVPrompt yields the best numbers (last row).

| Method | Vehicle(IoU=0.7) | | | Cyclist(IoU=0.5) | | | Pedestrian(IoU=0.5) | | |
|---|---|---|---|---|---|---|---|---|---|
| | Easy | Mid | Hard | Easy | Mid | Hard | Easy | Mid | Hard |
| BEVHeight w/ appearance-grouping (default) | 67.4 | 67.2 | 67.2 | 23.6 | 25.9 | 26.5 | 11.5 | 11.4 | 11.5 |
| BEVHeight w/ functionality-grouping (ours) | 72.0 | 73.6 | 72.0 | 23.5 | 25.7 | 26.4 | 10.1 | 10.1 | 10.2 |
| BEVPrompt w/ appearance-grouping | 70.5 | 71.4 | 71.3 | 27.6 | 31.3 | 31.6 | 15.2 | 15.2 | 15.2 |
| BEVPrompt w/ functionality-grouping | **72.8** | **74.2** | **72.5** | **27.7** | **31.5** | **31.8** | **15.3** | **15.2** | **15.2** |

Table 7: **Training pipeline in BEVPrompt.** When training the 3D detector of our BEVPrompt, we can either prompt ground-truth 2D boxes or predicted ones by the well-trained 2D detector. We compare them on on DAIR-V2X-I w.r.t AP in BEV. Results show that training the 3D detector with prompted 2D predictions outperforms that with prompted ground-truth 2D annotations.

| Method | Vehicle(IoU=0.7) | | | Cyclist(IoU=0.5) | | | Pedestrian(IoU=0.5) | | |
|---|---|---|---|---|---|---|---|---|---|
| | Easy | Mid | Hard | Easy | Mid | Hard | Easy | Mid | Hard |
| BEVHeight (as reference) | 67.4 | 67.2 | 67.2 | 23.6 | 25.9 | 26.5 | 11.5 | 11.4 | 11.5 |
| prompting 2D ground-truth | **72.8** | **74.2** | 72.5 | 27.7 | 31.5 | 31.8 | 15.3 | 15.2 | 15.2 |
| prompting 2D predictions | 72.5 | 74.0 | **74.1** | **29.8** | **31.5** | **31.8** | **16.4** | **16.3** | **16.3** |

pares these methods, showing that the our functionality-based grouping strategy indeed outperforms others. To justify this further, we train BEVHeight using our functionality-based grouping strategy, leading to remarkable improvement over the original version (cf. the first two rows in Table 6).

**How to train 3D detectors with prompted 2D detections.** Our BEVPrompt requires prompted 2D detections to make 3D predictions by its 3D detector. To train the 3D detector, we can either use ground-truth 2D boxes or predicted ones. We compare their performance w.r.t AP in BEV on the DAIR-V2X-I benchmark. Results in Table 7 show that directly prompting the predicted 2D boxes (with predicted class labels) performs better.

## 4.4 LIMITATIONS AND SOCIETAL IMPACTS

**Limitations.** While our method achieves the state-of-the-art roadside 3D detection performance on the existing benchmarks, it must output a 3D prediction for any prompted 2D detection. As 2D detections can be false positive, doing so inevitably hallucinates 3D detections which certainly are incorrect. Moreover, we note that the existing benchmarking protocol does not evaluate run-time. Our method requires more running time due to two visual encoders. Although running them can be in parallel, we hope to evaluate run-time or computation FLOPS in future benchmarking protocols.

**Societal impacts.** Although our method achieves significantly better 3D detection performance on the benchmark datasets, we note potential risks if directly using our method for vehicle-vehicle communication and vehicle-infrastructure cooperative perception. Real-world applications should treat our approach as a module and evaluate the whole system for the ultimate goals. Methods developed with existing datasets might be biased towards specific cities and countries. For example, in some countries, vehicles are running on the different side of the road from what are presented in the datasets. These are potential negative societal impacts.

## 5 CONCLUSION

We address the problem of roadside monocular 3D detection and propose a novel framework which leverages a well-trained 2D detector to improve the training of the 3D detector. Particularly, our method prompts and encodes predicted 2D boxes' $\{x, y, \text{width}, \text{height}, \text{label}\}$ as features, fuses such with 3D detector's features, and decode the fused features towards 3D detections. We validate our method through comprehensive experiments. Results show that our method significantly outperforms prior arts on publicly available benchmarks.

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

# Appendix

## A   FURTHER DETAILS IN PROMPT ENCODER AND FUSION MODULE

**Prompt encoder.**  A prompted 2D detection is mapped to a 512-dimensional embedding feature through three steps: 1) normalize the top-left and bottom-right coordinates (represented by $A \in \mathbb{R}^{2 \times 2}$) of a 2D detection box; 2) multiply $A$ by a random matrix $B \in \mathbb{R}^{2 \times 512}$ with Gaussian distribution, and add a learnable matrix $C \in \mathbb{R}^{2 \times 512}$ to get $D = A * B + C$, where $D \in \mathbb{R}^{2 \times 512}$; 3) encode the predicted class for the 2D detection as an index ID, repeat the ID value to a 512-dimensional vector, and concatenate this vector to $D$, resulting into $E \in \mathbb{R}^{3 \times 512}$, which is the final representation for the prompted detection and input to the fusion module detailed next.

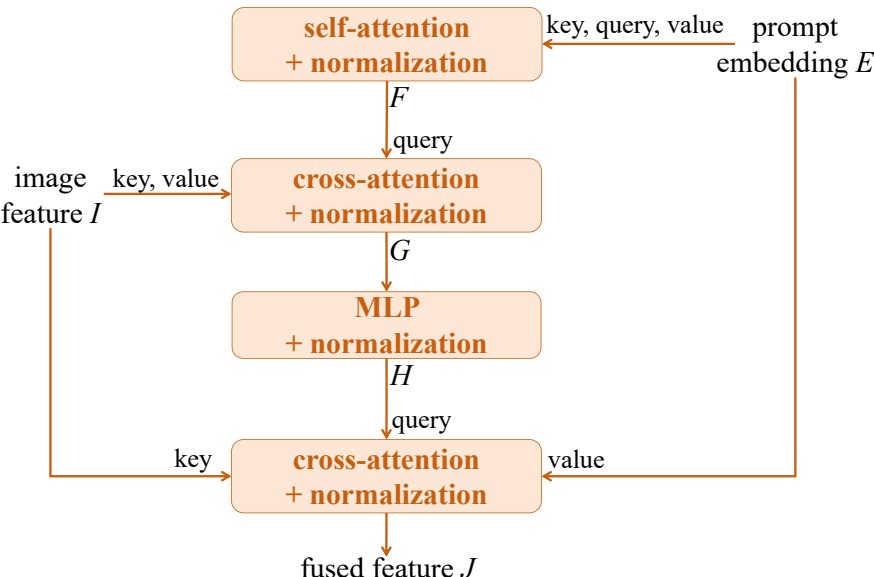

Figure 4: Illustration of our proposed fusion module, which takes the prompted feature $E$ from 2D detector and image feature $I$ from 3D detector as input and outputs the fused feature $J$.

**Fusion module** aims to fuse prompted feature $E$ with image feature $I$ encoded by the 3D detector encoder. To fuse them, we use a transformer module. Specifically, we use a 2-layer decoder, as illustrated in Fig. 4. The fusion module has four steps of computation: 1) applying self-attention on prompt embedding $E$ and normalizing it to obtain $F$; 2) applying cross-attention on $F$ and normalizing it to obtain $G$, where key and value of cross-attention are the image feature $I$ and query is $F$; 3) updating $G$ through point-wise MLP and normalizing it to obtain $H$; 4) applying cross-attention on $H$ and normalizing it to obtain $J$, where $J$ is the output of fusion module. The shape of $J$ is the same as the image features $I$. The key, query, value of cross-attention are $I$, $H$, $E$, respectively.

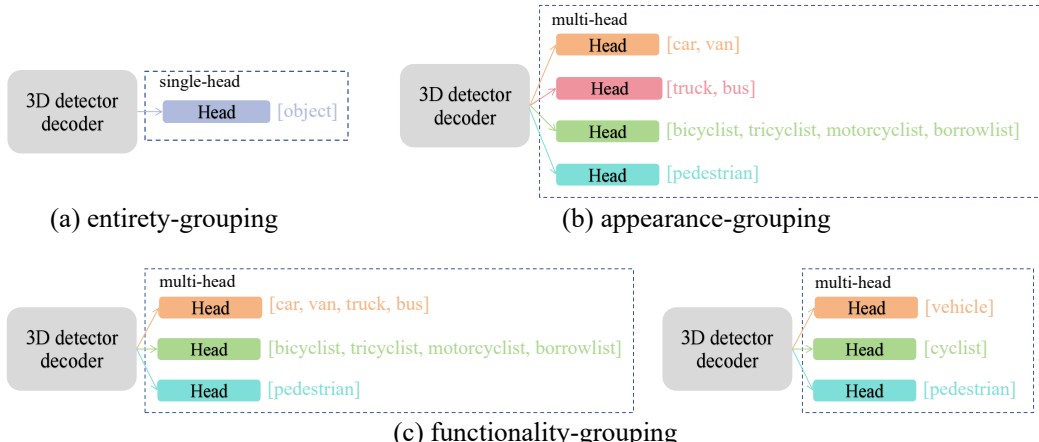

Figure 5: Graphic illustration of grouping strategies and detector head designs. The different grouping methods used in detection head in the 3D detector include entirety-grouping, appearance-grouping, functionality-grouping. Design (a) merges all the classes into a single super-class "object" with single-head. Predicted labels for detections are given by the 2D detector. Design (b) groups classes according to their appearance similarity with multi-head. Design (c) groups classes according to their functionalities with multi-head. We can use either fine-grained classes in a head or super classes in a head showing on the left and right respectively.

## B  FURTHER ANALYSIS OF CLASS-GROUPING STRATEGIES

While data is oftentimes labeled with a vocabulary of fine-grained classes, real-world applications might not require distinguishing some fine-classes but consider them as a whole. For example, the DAIR-V2X-I dataset annotates images w.r.t {*truck*, *bus*, *car*, *van*, *bicyclist*, *tricyclist*, *motorcyclist*, *barrowlist*, *pedestrian* }, but its evaluation protocol reports numbers on superclasses including {*vehicle*, *cyclist*, *pedestrian*}, where *vehicle*={*truck*, *bus*, *car*, *van*}, and *cyclist*={*bicyclist*, *tricyclist*, *motorcyclist*, *barrowlist*}.

On the other hand, to facilitate training, a common practice is to group fine-classes into superclasses and train superclass-specific heads, each of which makes predictions only within the specific superclass. Class grouping is oftentimes based on object appearance. For example, BEVHeight adopts an appearance-based grouping strategy, creating three superclasses that merge {*truck*, *bus*}, {*car*, *van*}, and {*bicyclist*, *tricyclist*, *motorcyclist*, *barrowlist*}, respectively, along with the origin *pedestrian* class. As we can see, both *truck* and *bus* are big vehicles so they are grouped together. Based on such superclasses, BEVHeight learns a four-head detector.

In this paper, we advocate a functionality-based grouping strategy, motivated by the fact that vehicles, cyclists and pedestrians appear at relatively fixed regions on the image. We carry out comprehensive analysis how these grouping strategies affect the final roadside 3D detection performance.

Note that 2D detectors can also adopt class grouping, and for each detector head, we can train it using a single superclass label (aka one-way classifier) or multiple fine-class labels (aka $K$-way classifier). We combine all these different class grouping and multi-head designs in both 2D detector and 3D detector in this study. We use the DAIR-V2X-I dataset in this study. Fig. 5 illustrates detector-head designs and Table 8 lists quantitative comparisons. We summarize key conclusions below.

- Comparing the first two rows in Table 6 shows that functionality-grouping yields much better performance than appearance-based grouping (which is the default strategy in BEVHeight).

- Comparing results within the 2nd or 3rd row-panel , we see it does not make a difference whether to use fine-class labels or super-class labels to train 2D detectors. This suggests that training 2D detectors is quite robust to such configurations.

- Comparing results across the 2nd and 3rd row-panel, we see that multi-head 3D detector plus $K$-way classifier (i.e., each head makes $K$-way classification on its fine-classes) per-

forms much better than single-head design — the former achieves 15.2 AP whereas the latter achieves 11.8.

- Within the last row-panel, we see that with functionality-based grouping, training a multi-head 3D detector using superclass labels only is better than using fine-class labels (i.e., more fine-class labels within each superclass for each detector head).

Table 8: **Results of different class-grouping strategies and detector-head designs.** In addition to the studies in Table 5, we compare *entirety-based grouping* which merges all classes together into a single class, using predicted labels from the 2D detector as the label in 3D detections. For functionality-based grouping, we study training with fine-classes in a head vs. super-class in a head (cf. Fig.5c), where the latter yields the best results.

| Method | Vehicle$_{(IoU=0.7)}$ | | | Cyclist$_{(IoU=0.5)}$ | | | Pedestrian$_{(IoU=0.5)}$ | | |
|---|---|---|---|---|---|---|---|---|---|
| | Easy | Mid | Hard | Easy | Mid | Hard | Easy | Mid | Hard |
| BEVHeight w/ appearance-grouping, multi-head (default) | 67.4 | 67.2 | 67.2 | 23.6 | 25.9 | 26.5 | 11.5 | 11.4 | 11.5 |
| BEVHeight w/ functionality-grouping, multi-head (ours) | 72.0 | 73.6 | 72.0 | 23.5 | 25.7 | 26.4 | 10.1 | 10.1 | 10.2 |
| BEVPrompt w/ entirety-grouping, single-head, one-way cls + DINO w/ super-class cls | 71.6 | 71.0 | 71.0 | 26.7 | 31.1 | 31.3 | 11.9 | 11.8 | 11.9 |
| BEVPrompt w/ entirety-grouping, single-head, one-way cls + DINO w/ fine-class cls | 71.3 | 71.3 | 71.1 | 26.6 | 30.0 | 31.3 | 11.8 | 11.8 | 11.8 |
| BEVPrompt w/ appearance-grouping, multi-head, $K$-way cls + DINO w/ super-class cls | 70.5 | 71.4 | 71.3 | 27.6 | 31.3 | 31.6 | 15.2 | 15.2 | 15.2 |
| BEVPrompt w/ appearance-grouping, multi-head, $K$-way cls + DINO w/ fine-class cls | 70.0 | 71.6 | 71.3 | 27.4 | 31.4 | 31.2 | 15.3 | 15.1 | 15.2 |
| BEVPrompt w/ functionality-grouping, multi-head, $K$-way cls + DINO w/ super-class cls | 71.9 | 73.0 | 71.9 | 27.5 | 31.5 | 31.6 | 15.1 | 15.2 | 15.0 |
| BEVPrompt w/ functionality-grouping, multi-head, $K$-way cls + DINO w/ fine-class cls | 71.7 | 72.9 | 71.9 | 27.4 | 31.3 | 31.4 | 15.0 | 15.0 | 15.1 |
| BEVPrompt w/ functionality-grouping, multi-head, one-way cls + DINO w/ super-class cls | **72.8** | **74.2** | 72.5 | **27.7** | **31.5** | **31.8** | **15.3** | **15.2** | **15.2** |
| BEVPrompt w/ functionality-grouping, multi-head, one-way cls + DINO w/ fine-class cls | 72.1 | 74.2 | **72.6** | 27.5 | 31.7 | 31.4 | 15.1 | 15.0 | 15.0 |

## C IMPROVING 3D DETECTION BY TUNING YAW

3D cuboids predicted by a 3D detector might not well align with the 3D objects in terms of orientation. Fig. 6 shows this issue on the vehicle class. Because of this, projecting 3D cuboids on the image plane are likely to be misaligned with 2D boxes predicted by the 2D detector, which tends to performs sufficiently well for 2D detection. This motivates us to optimize the orientation for each 3D detection using the corresponding 2D detection. Specifically, We do so by rotating a 3D cuboid to maximize the IoU of its 2D projected box with the corresponding detection box by the 2D detector. Table 9 compares results with and without yaw tuning, demonstrating that tuning yaw improves vehicle detection by 0.2 mAP. Fig. 6 shows visual comparisons.

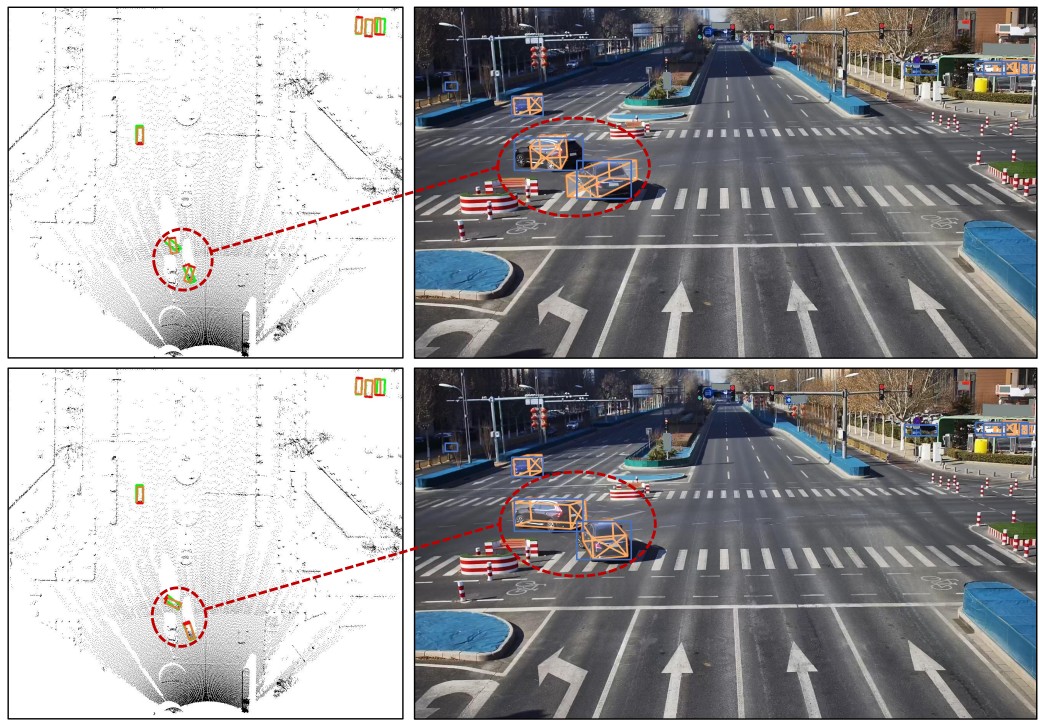

Figure 6: Visual comparison of with (upper panel) and without (lower panel) yaw tuning. We visualize 3D detection on the BEV plane (left) and image plane (right). Ground-truth and detection boxes are in green and orange on the BEV plane, respectively. We visualize 3D and 2D detection boxes in orange and blue on the image plane, respectively. Tuning yaw clearly improves 3D detection performance.

Table 9: **Tuning yaw for improves 3D detection performance.** For the vehicle class, we tune the yaw/orientation of 3D detections by BEVPrompt by exploiting 2D detections. This is motivated that 3D vehicle detections do not necessarily have good orientations. When projected on the image plane, 3D cuboids do not align well with 2D boxes of the 2D detector. We optimize yaw by rotating 3D cuboid to maximize its IoU with the corresponding 2D detection.

| Method | Vehicle$_{(IoU=0.7)}$ | | |
|---|---|---|---|
| | Easy | Mid | Hard |
| BEVHeight | 67.4 | 67.2 | 67.2 |
| No tune yaws | 72.5 | 74.0 | 74.1 |
| Tune yaws | **72.7** | **74.3** | **74.3** |

# D    MORE VISUALIZATION

Figure 7 visualizes more results on the DAIR-V2X-I dataset (Yu et al., 2022). BEVPrompt not only has better orientation predictions for vehicles, but also can recall infrequently-seen ambulances, occluded pedestrians and cyclists.

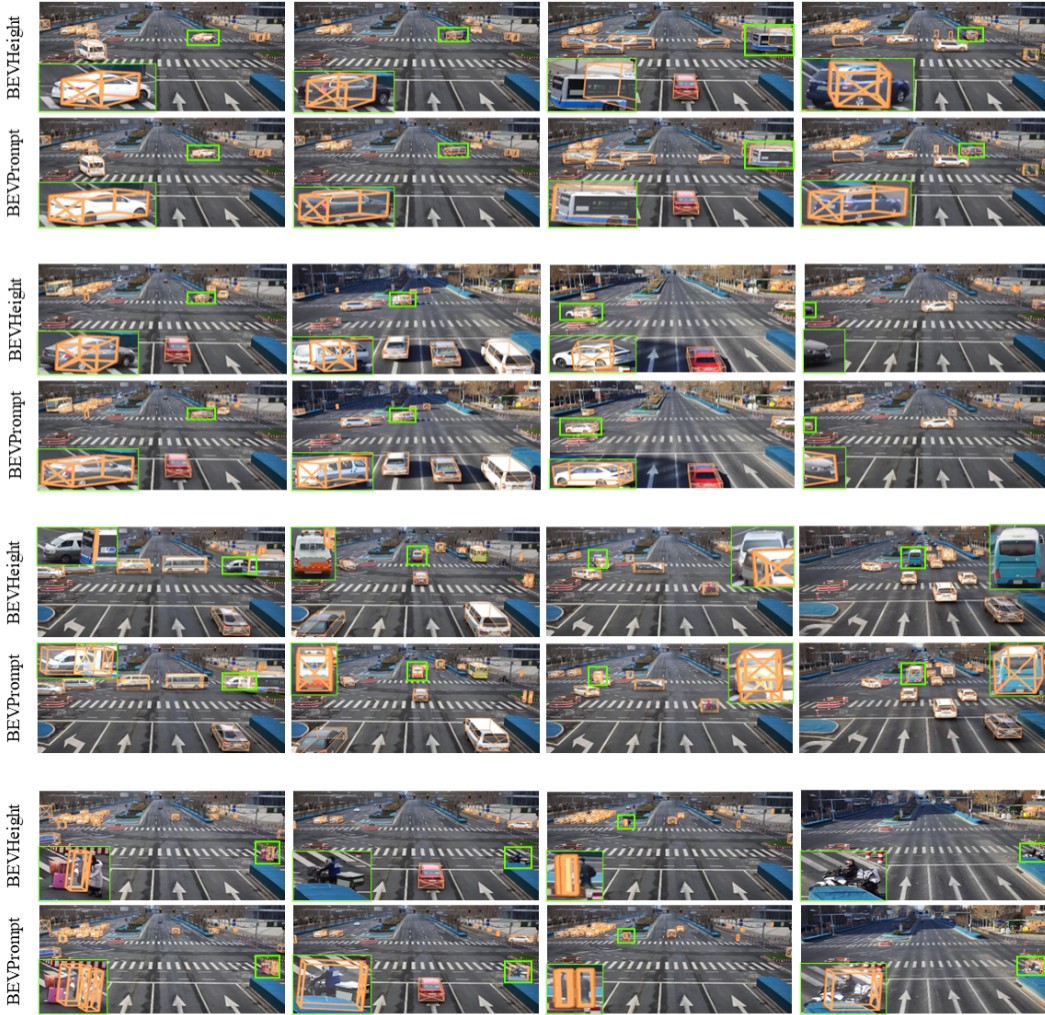

Figure 7: More visualizations between the state-of-the-art method BEVHeight (Yang et al., 2023) and our BEVPrompt.

