# OpenReview forum: "Roadside Monocular 3D Detection via 2D-Detection Prompting"
_ICLR.cc/2024/Conference — ICLR 2024 Conference Withdrawn Submission_

### Official Review · Reviewer_n9u2 · 2023-10-25

**Soundness:** 2 fair
**Presentation:** 3 good
**Contribution:** 2 fair
**Rating:** 5
**Confidence:** 4

**Summary:**

The paper proposes BEVPrompt, a query-based detector for outputting 3D bounding boxes for roadside monocular 3D detection tasks. The method relies on the well-studied 2D detection first. Then, it uses attention to fuse the 2D boxes and 3D features with the box parameters serving as queries to the output 3D boxes. Experiments on roadside benchmarks such as DAIR-V2X-I and Rope3D show effectiveness.

**Strengths:**

+ The paper uses the 2D bounding box parameters as a query, the first application in the roadside perception task.
+ The writing is good and easy to understand.
+ The paper achieves great results on the two benchmark datasets.

**Weaknesses:**

- BEVPrompt does not compare with query-based methods such as SparseBEV [A], CAPE or PETR, which uses queries and/or pillars to get 3D boxes. Since the sparseBEV code is available, we should re-train this baseline on the DAIR-V2X and Rope3D datasets. These comparisons are important since the authors admit that the 2D bounding box parameters work the best.

- The idea of why using all the 2D box parameters works better than the 2D features needs to be justified and rigorously proved. In other words, can the authors provide a solid mathematical/analytical proof of why these empirical results make sense?

- It would also be good to quantitatively ablate all the possible combinations of the bounding box as queries:
  - Only x,
  - Only y,
  - Only label,
  - h+w,
  - x+label,
  - y+label,
  - x+y+label,
  - x+h+w,
  - y+h+w,
  - x+h+w+label,
  - y+h+w+label

- Since the method uses 2D detections, my hunch is that there would be truncation of the partially visible boxes (say on the sides of the image). Therefore, I would ask the authors to manually select at least 500 truncated images in the Val set and quantitatively report the performance of all the baselines (especially MonoFlex, BEVDepth, and BEVHeight) and BEVPrompt in Table 2 of the paper.

- Mono3D task results, in general, are susceptible to seed selection. The authors do a validation checkpoint selection, which makes their results irreproducible. Please run the models on five seeds as in DEVIANT [B] and report the median model and standard deviation obtained at the end of the last epoch. In other words, I want to see the results without validation checkpoint selection and the authors should report the median performance of the five seeded models at the end of the last epoch.

References:
- [A] SparseBEV, Liu et al., ICCV 2023.
- [B] DEVIANT, Kumar et al., ECCV 2022.

**Questions:**

Please see the weakness section.

---

### Official Review · Reviewer_QPjq · 2023-10-31

**Soundness:** 2 fair
**Presentation:** 2 fair
**Contribution:** 2 fair
**Rating:** 5
**Confidence:** 5

**Summary:**

This work exploits 2D detection results to help roadside monocular 3D Detections. The authors propose a new method called BEVPrompt to fuse 3D features with prompted 2D detection results and yield significant performance improvement. At the same time, they also designed a new class grouping mechanism to boost the final detection further. The proposed method establishes new SotA results on both DAIR-V2X-I and Rope3D benchmarks.

**Strengths:**

- Prompting 2D detection results to help 3D detection is interesting.
- Proposed functionality-based grouping is straightforward yet effective.

**Weaknesses:**

- Lack of detailed discussion in the ablation study section. For instance, we don't know the true reason why 2D prompting is helping roadside 3D detection. Is it the classification? Is it the localization?
- The main paper has some duplicated content, which leaves no space for something important like your design of the prompting model.
- The results with different 2D detectors are missing. Not sure whether the main performance improvement is from DINO itself.

**Questions:**

- The claiming of using multiple losses to train 2D and 3D detectors together might lead to the missing of some hard cases as an attempt to handle the difficulty in regressing coordinates, depth, orientation... etc. Are there any experimental results to support it?
- Since the 2D annotation is easier to obtain, are there any results that support that training the 2D detector with more data can further improve roadside 3D detection?

---

### Official Review · Reviewer_pSM9 · 2023-11-02

**Soundness:** 2 fair
**Presentation:** 1 poor
**Contribution:** 1 poor
**Rating:** 3
**Confidence:** 5

**Summary:**

The paper tackles the problem of roadside monocular 3D object detection. The authors propose a new method called BEVPrompt that leverages a pretrained 2D detector to help train the 3D detector. The motivation is that 2D detection is an easier task and there are more datasets available for pretraining good 2D detectors. BEVPrompt has two main components: a prompt encoder that encodes the 2D detections into feature vectors, and a fusion module that fuses the 2D detection prompts with the 3D detector's own features. The authors further improve performance by using functionality-based class grouping when training the multi-head 3D detector. Experiments on two roadside 3D detection benchmarks (DAIR-V2X-I and Rope3D) show the final solution outperforms prior state-of-the-art methods.

**Strengths:**

1. The proposed solution improves over BEVHeight on two road side monocular 3D detection datasets.

**Weaknesses:**

1. The topic of this work is too narrow for ICLR. It focuses on improving road side monocular 3d object detection with results from 2d detectors. The authors only reported results on two road side monocular 3d datasets. 3d object detection for road side perception is not a new breakthrough on the application side.
2. The writing of the paper is somewhat problematic. In Sec. 3.1, the authors listed 3 different types of prompts could be provided by a 2d detector and 3 different fusion strategies. An academic paper is not an experiment report, listing a bunch of very different design make this work hard to comprehend.
3. The structure of this paper is also very strange. I don’t know why the authors decide to put the main illustration of their method in the appendix, which is supposed to the most important part of this paper. The authors instead spend a lot of space in Sec 3.2 for repeating what is already stated in the introduction and in Sec 3.3 for introducing other people’s work.
4. This work did not proper discuss the relation and difference between 2D object proposal and the so-called “2d box prompt”.
5. The novelty of this work is quite limited as using embeddings derived from 2d box as the query for object decoder in DETR-like architecture is well explored by in both 2D OD by DBA-DETR / DINO and 3D OD by PETR and BEVFormer.

**Questions:**

1. The authors may include more diverse datasets for benchmarking the proposed method. Like adding results on KITTI and nuScenes to engage researchers in the autonomous driving field or adding results on NYUv2 and ScanNet to engage researchers in the home robotics, AR/VR/MR and indoor SLAM fields. If adding results for a more broad audience is not possible, the authors may consider to resubmit this work to traffic-related venues like IEEE IV to target the right audience.

---

### Official Review · Reviewer_XGju · 2023-11-03

**Soundness:** 3 good
**Presentation:** 3 good
**Contribution:** 3 good
**Rating:** 6
**Confidence:** 4

**Summary:**

This paper proposes to use the output of a 2D detector to improve monocular 3D object detection. It first trains a 2D detector to generate 2D bounding boxes of objects in images of a roadside dataset. The box information is then used as prompts for an off-the-shelf 3D detector. Experiments on DAIR-V2X-I and Rope3D show improvement over the baseline that does not use such prompts.

**Strengths:**

- The idea is clear and easy to understand. The text and figures provide an adequate explanation of the proposed method.
- The method is properly motivated with the experiment in Figure 1 (b). In this experiment, the authors show that the state-of-the-art 3D detector for the dataset at hand underperforms a common  2D detector in terms of the quality of the 2D bounding boxes (projected from 3D).
- Experiments on two datasets show consistent improvement over the baseline method (BEVHeight).
- Ablation results are adequate.

**Weaknesses:**

- The paper does not include a study on the additional runtime required by the 2D detector (mentioned in the limitations). The 2D detector increases the computation requirement for training and inference. Moreover, the features are extracted independently for the 2D and 3D detectors which might be learning redundant information.
- The method is tested on roadside scenarios (DAIR-V2X-I and Rope3D), but not on ego-vehicle settings such as in the nuScenes dataset.
- The results are shown on the validation set, which raises concerns of overfitting.
- The training of the final pipeline is not end-to-end. It requires a training of a 2D detector and is affected by the quality/difference in quality of detections during training and inference time.

**Questions:**

- How does the proposed method perform on monocular 3D object detection on nuScenes?
- How does the runtime compare to other baseline methods?

---

### Public Comment · ~Zehao_Huang1 · 2023-11-13
**Suggestion for Additional Reference in MV2D**

Dear authors, I have read your paper and found it very insightful. I couldn't help but notice the similar motivation between your work and our research MV2D [1] published in ICCV2023. Our paper consider adopting 2D object detection to enhance the performances of multi-view 3D detection. I believe that referencing our paper in your work could provide additional context to your findings, potentially benefiting readers who are interested in this subject area. I thought it might have been an oversight, and I wanted to bring it to your attention.

Could you mind adding our paper to your reference in your next revision? I appreciate your consideration of this suggestion. Thanks!

[1] Wang, Zitian, et al. "Object as Query: Lifting Any 2D Object Detector to 3D Detection." Proceedings of the IEEE/CVF International Conference on Computer Vision. 2023.